# Pressure Effects with Incorporated Particle Size Dependency in Graphene Oxide Layers through Observing Spin Crossover Temperature

**Hikaru Kitayama [1], Ryohei Akiyoshi [1], Masaaki Nakamura [1] and Shinya Hayami [1,2,*]**

[1] Department of Chemistry, Graduate School of Science and Technology, Kumamoto University, 2-39-1 Kurokami, Chuo-ku, Kumamoto 860-8555, Japan; 179d8024@st.kumamoto-u.ac.jp (H.K.); 187d9041@st.kumamoto-u.ac.jp (R.A.); m_nakamura@kumamoto-u.ac.jp (M.N.)

[2] Institute of Pulsed Power Science (IPPS), Kumamoto University, 2-39-1 Kurokami, Chuo-ku, Kumamoto 860-8555, Japan

[*] Correspondence: hayami@kumamoto-u.ac.jp; Tel.: +81-096-342-3469

**Abstract:** This research highlights the pressure effects with the particle size dependency incorporated in two-dimensional graphene oxide (GO)/reduced graphene oxide (rGO). GO and rGO composites employing nanorods (NRs) of type $[Fe(Htrz)_2(trz)](BF_4)$ have been prepared, and their pressure effects in the interlayer spaces through observing the changes of the spin crossover (SCO) temperature ($T_{1/2}$) have been discussed. The composites show the decrease of interlayer spaces from 8.7 Å to 3.5 Å that is associated with GO to rGO transformation. The shorter interlayer spaces were induced by pressure effects, resulting in the increment of $T_{1/2}$ from 357 K to 364 K. The pressure effects in the interlayers spaces estimated from the $T_{1/2}$ value correspond to 24 MPa in pristine $[Fe(Htrz)_2(trz)](BF_4)$ NRs under hydrostatic pressure. The pressure observed in the composites incorporating NRs ($30 \times 200$ nm) is smaller than that observed in the composite incorporating nanoparticles (NPs) (30 nm). These results clearly demonstrated that the incorporated particle size and shape influenced the pressure effects between the GO/rGO layer.

**Keywords:** pressure effect; spin crossover; graphene oxide; iron complex

## 1. Introduction

Van der Waals interactions in the pores of micro-porous materials are known to generate a pseudo-pressure effect, leading to the expression of characteristic phases and unique properties in the pores under mild conditions. [1–6]. For example, potassium iodide (KI) nanocrystals inside the nanotube spaces of single-walled carbon nano-horns display a structural phase transition by pseudo-pressure corresponding to *ca.* 1.9 GPa [7]. In microporous of metal-organic frameworks $[\{[Cu_2(pzdc)_2(pyz)]\cdot 2H_2O\}_n]$ (pzdc—pyrazine-2,3-dicarboxylate), $O_2$ molecules show similar behavior to the solid phase above the freezing point of $O_2$ [8].

Recently, interlayers of two-dimensional (2D) materials, such as graphene, boron nitride (BN), and $MoS_2$, were found to play an important role for the confinement of molecules and pseudo-pressure effects [9–12]. For instance, pressure corresponding to 1.2 ± 0.3 GPa was observed by trapping pressure-sensitive molecules of triphenyl amine (TPA) and boric acid (BA) into an interlayer of graphene [9]. In typical 2D layered materials, the correlations between pressure ($P$) and the interlayer distance ($d$) were estimated using the equation of $P \approx E_w/d$, where $E_w$ is the adhesion energy [11,12]. As such this is an indication that the pressure effects that occur in the interlayer are significantly affected by the interlayer distance. Thus, 2D materials that possess a tunable interlayer have the possibility of tuning pressure effects, leading to the generation of unique phases and physical properties.

Graphene oxide (GO), an oxidized graphene, is a 2D material that has oxygen functional groups, such as hydroxyl, carboxyl, and epoxy groups [13–16]. These oxygen functional groups on the GO surface were removed by thermal reduction treatment, resulting in reduced graphene oxide (rGO) [17–19]. Importantly, their interlayer distances decrease from 7.9 Å in GO to 3.4 Å in rGO as a result of the removal of the oxygen functional groups [20–22]. Therefore, a pseudo pressure effect can be generated via GO/rGO transformation.

Recently, we reported tunable pressure effects on GO/rGO layers by changing the thermal treatment temperature [23]. In this context, nanoparticles (NPs) of a spin crossover (SCO) complex of type $[Fe(Htrz)_2(trz)](BF_4)$ (trz: 1,2,4-triazole) were used for monitoring the pressure effect changes on the GO/rGO layers. SCO complexes, $[Fe(Htrz)_2(trz)](BF_4)$, are also well known to exhibit SCO phenomena between low spin (LS) and high spin (HS) states, reversibly with thermal hysteresis [24–27]. In addition, the SCO temperature ($T_{1/2}$) of $[Fe(Htrz)_2(trz)](BF_4)$ is sensitive to the hydrostatic pressure that behaves to restrict the structural transition synchronized with the SCO behavior. The correlation between $P$ and $T_{1/2}$ has also been reported in $[Fe(Htrz)_2(trz)](BF_4)$ NPs [28]. In a prior study, we reported that the composite incorporating $[Fe(Htrz)_2(trz)](BF_4)$ bulk particles (BPs) of 100 nm did not show any pressure effect, but did exhibit a pressure effect for the composite incorporating NPs of 30 nm, since the GO nanosheet can cover the NPs completely.

In the present study, we aimed to further investigate the pressure effects between the GO/rGO layers. For this purpose, we prepared GO (**1**)/rGO (**2**) composites incorporating cylinder shape nanorods (NRs) $[Fe(Htrz)_2(trz)](BF_4)$ with a size of $30 \times 200$ nm (intermediate size of previously reported particle) as a way to detect the pressure effects (Figure 1). Then, pressure effects in the GO/rGO layers were discussed by monitoring $T_{1/2}$, and a comparison was made with GO/rGO composites incorporating spherical NPs ($30 \times 30$ nm) and bulk particles ($100 \times 100$ nm) respectively.

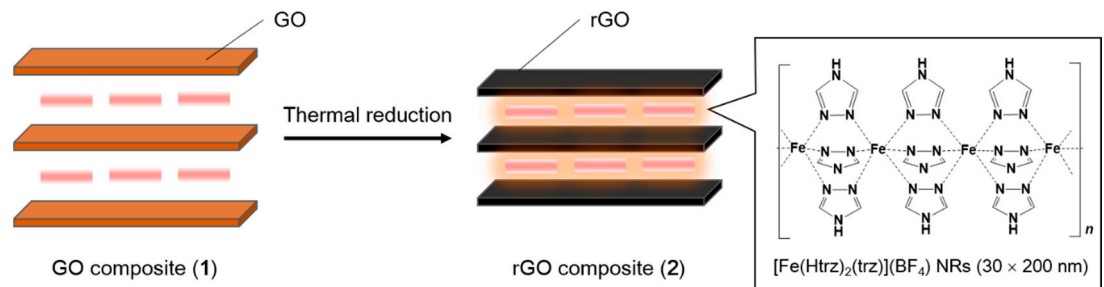

**Figure 1.** Schematic illustration of pressure effects in graphene oxide (GO)/ reduced graphene oxide (rGO) layers incorporating $[Fe(Htrz)_2(trz)](BF_4)$ nanorods (NRs).

## 2. Results and Discussion

The $[Fe(Htrz)_2(trz)](BF_4)$ NRs were synthesized by the reaction between $FeCl_2 \cdot 4H_2O$, $NaBF_4$, and 1-H-1,2,4-triazole, using the ligand-melt method [29]. Composite **1** was prepared by mixing GO and $[Fe(Htrz)_2(trz)](BF_4)$ NRs in a mass ratio of 1:2 in ethanol, which was then filtrated. Composite **2** was obtained by subsequent heating at 473 K for 12 h. The GO/rGO transformation in these composites was confirmed by investigating the current–voltage (I–V) properties. The I–V curve for composite **1** shows mainly an insulator property in accordance with the behavior of GO. The electron conductivity of composite **1** was $7.67 \times 10^{-11}$ A, applied at 1 V. On the other hand, composite **2** showed $7.28 \times 10^{-6}$ A applied at 1 V, in accordance with the oxygen functional groups being removed to yield rGO. This transformation is also corroborated by the powder X-ray diffraction (PXRD) patterns, as presented in Figure 3.

The scanning electron microscopy (SEM) images of the $[Fe(Htrz)_2(trz)](BF_4)$ NRs, composite **1**, and composite **2** are presented in Figure 2 and Figure S2. The SEM image demonstrated that the size of the NR complex was 29.6 nm in width and 203.4 nm in length. For composites **1** and **2**, the NRs incorporated between the GO/rGO layers were observed obviously. Furthermore, the presence

of [Fe(Htrz)$_2$(trz)](BF$_4$) was clearly confirmed by the energy dispersive X-ray (EDX) spectroscopy (Figure 2c,d). The Fourier transform infrared spectra (FT-IR) results also supported the presence of [Fe(Htrz)$_2$(trz)](BF$_4$) NRs composited within the GO/rGO interlayers (Figure S3).

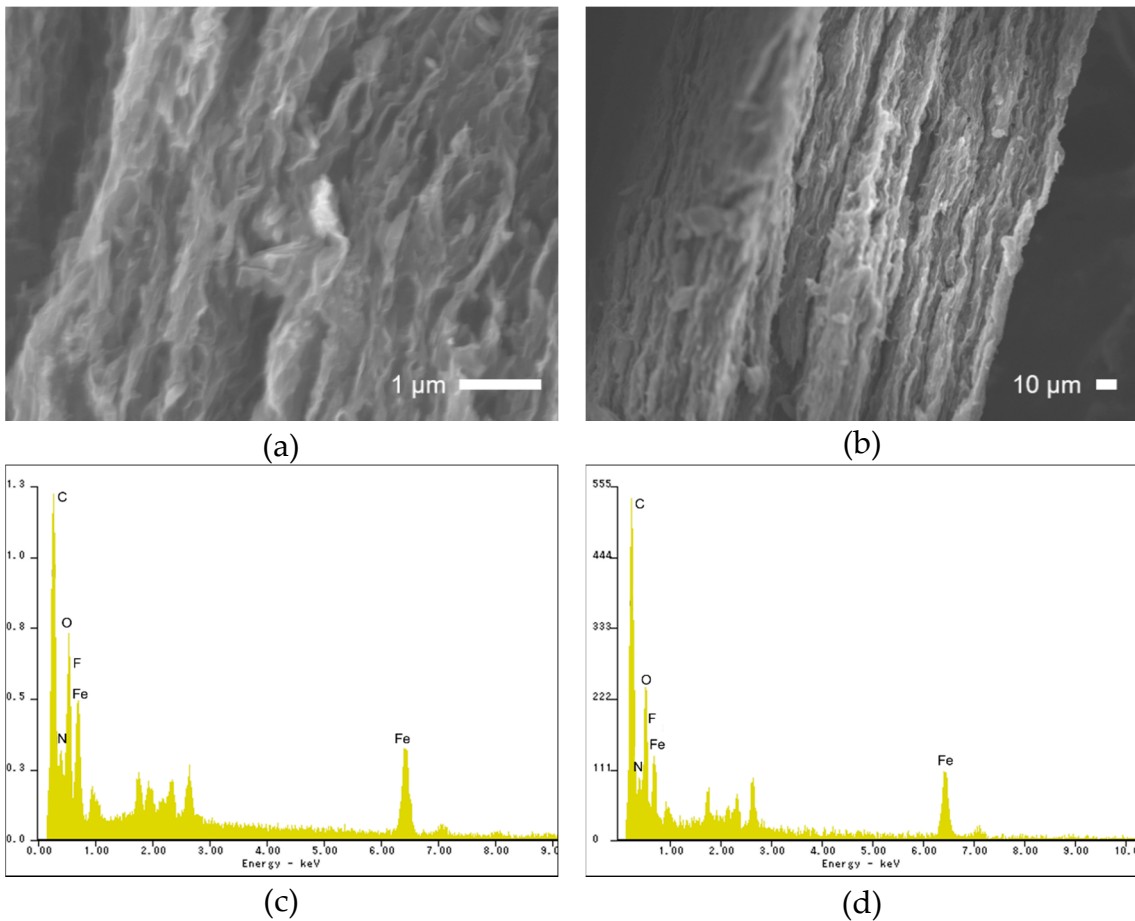

**Figure 2.** Scanning electron microscopy (SEM) images of (**a**) composite **1** and (**b**) composite **2**. SEM-energy dispersive X-ray spectroscopy (SEM-EDX) results for (**c**) composite **1** and (**d**) composite **2**.

As such, the changes of the interlayer distance that is associated with the transformation of GO to rGO were investigated by powder X-ray diffraction (PXRD) measurements (Figure 3). Results shows that pristine GO has a distinct peak at $2\theta = 10.15°$, with an interlayer distance of 8.70 Å. As for composite **1**, the GO peak was observed at $2\theta = 10.17°$ and an interlayer distance of 8.68 Å, where the remaining peaks are ascribed to the presence of [Fe(Htrz)$_2$(trz)](BF$_4$) NRs. In the case of composite **2** (which was treated at 473 K for 12 h), the interlayer distance decreased to 3.5 Å ($2\theta = 25°$) as a result of the removal of the oxygen functional groups on the GO layers. From these results, it can be anticipated that pressure effects occurred between the interlayers.

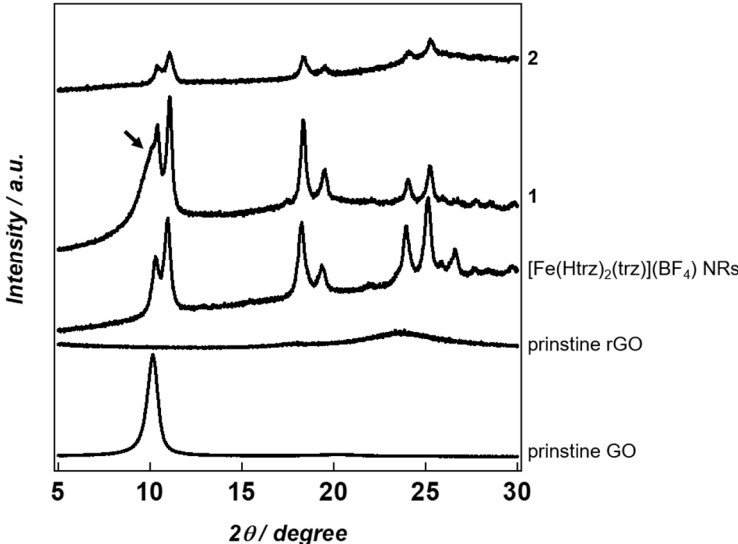

**Figure 3.** Powder X-ray diffraction (PXRD) patterns for pristine GO, pristine rGO, $[Fe(Htrz)_2(trz)](BF_4)$ NRs, composite **1**, and composite **2**.

In order to investigate the influence of the pressure effects on the SCO behavior, caused by the shorter interlayer distance associated with the structural transformation between GO and rGO, the temperature-dependent magnetic susceptibility for the $[Fe(Htrz)_2(trz)](BF_4)$ NRs, composite **1**, and composite **2** were measured in the temperature range of 300 to 400 K. The magnetic susceptibility for the $[Fe(Htrz)_2(trz)](BF_4)$ NRs in the form of the $\chi_m T$ vs. $T$ plot can be seen in Figure S4, where $\chi_m$ is the molar magnetic susceptibility and $T$ is the temperature. From these results, $[Fe(Htrz)_2(trz)](BF_4)$ NRs show SCO behavior at $T_{1/2} = 356$ K, with a thermal hysteresis of 29 K. The $\chi_g T$ vs. $T$ plots for composite **1** and composite **2** are shown in Figure 4, where $\chi_g$ is the magnetic susceptibility per gram. Both composites **1** and **2** exhibited SCO behavior at $T_{1/2} = 357$ K and 364 K respectively. The $T_{1/2}$ value of composite **2** is 7 K higher than that observed in composite **1**. Accordingly, these results are in agreement with pressure effects behavior when decreasing the interlayer distance.

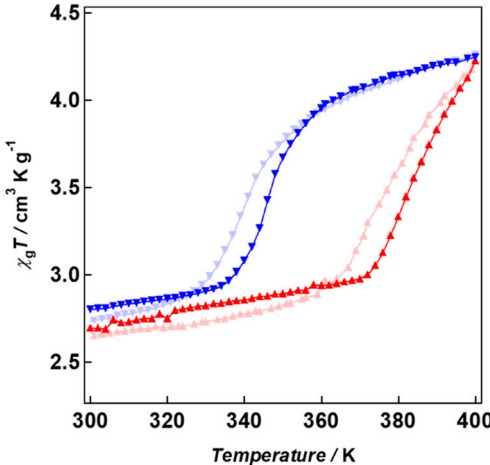

**Figure 4.** $\chi_g T$ vs. $T$ plots for composite **1** (heating: (△); cooling: (▽)) and composite **2** (heating: (▲); cooling: (▼)).

The pseudo-pressure effects were estimated from the $T_{1/2}$ value using the Clausius–Clapeyron equation (Equation (1)) reported by Colacio and co-workers, where $p$ is hydrostatic pressures [28], as follows:

$$T_{1/2}(p) = T_{1/2} + 290(66)p \tag{1}$$

The values of SCO temperature and pseudo-pressure for GO/rGO composite when [Fe(Htrz)$_2$(trz)](BF$_4$) of different size and shape are incorporated are summarized in Table 1. As a result of the calculation, the pseudo-pressure originated from the transformation of composite **1** to composite **2** is equal to 24 MPa. We have reported previously that GO/rGO composites incorporating [Fe(Htrz)$_2$(trz)](BF$_4$) NPs with a size of 30 nm show an increase of the $T_{1/2}$ value from $T_{1/2}$ = 351 K in the GO, to $T_{1/2}$ = 362 K in rGO due to pressure effects corresponding to 38 MPa [23]. The pseudo-pressure effect observed in the composite with NRs (30 × 200 nm) was smaller than that observed in the composite with NPs (30 nm). Considering that no pressure effects were observed for the composite incorporating BPs of 100 nm size, it can be concluded that the accommodated particle size and shape crucially affected the pseudo-pressure effects within the GO/rGO layers. For the case of small particle size, the GO layers stack regularly. GO layers form the ordered stacking structures when incorporating NRs, however, the surface area of the NRs influencing the pressure effects is larger than the NPs with a size of 30 nm (Figure 5). It is then proposed that a large surface of NRs leads to small pressure effects.

**Table 1.** Summary of spin crossover (SCO) temperatures ($T_{1/2}$) and pseudo-pressure.

|  | $T_{1/2}$ **(K)** | **Pseudo-Pressure** |
|---|---|---|
| **1** (GO/NRs) | 357 | 24 MPa |
| **2** (rGO/NRs) | 364 | |
| GO/NPs | 351 | 38 MPa |
| rGO/NPs | 362 | |
| GO/BPs | 357 | No pressure |
| rGO/BPs | 352 | |

The particle size is 30 × 200 nm in nanorods (NRs), 30 nm in nanoparticles (NPs), and 100 nm in bulk particles (BPs).

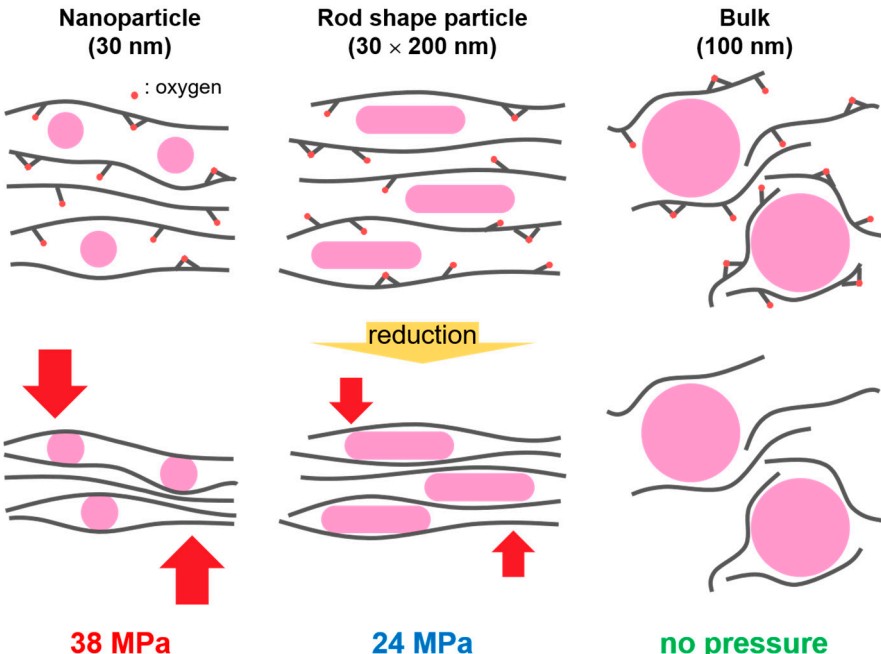

**Figure 5.** Schematic illustration of the pressure effects with incorporated particle size and shape dependency in the GO/rGO layers.

## 3. Materials and Methods

### 3.1. Synthesis

All the materials and reagents were obtained from Wako Pure Chemical Industries (Osaka-shi, Osaka, Japan) and Tokyo Chemical Industry (TCI) Co., Ltd (Chuo-ku, Tokyo, Japan) and used without further purification.

#### 3.1.1. [Fe(Htrz)$_2$(trz)](BF$_4$) NRs (30 × 200 nm)

The [Fe(Htrz)$_2$(trz)](BF$_4$) NRs were prepared according to the previously reported procedure [29]. The mixture of FeCl$_2$·4H$_2$O (200 mg, 1 mmol), NaBF$_4$ (110 mg, 1 mmol), and 1-H-1,2,4-triazole (5 g, 72.4 mmol) was heated at 150 °C for 5 min. After heating, the resulting melt was cooled to room temperature. The obtained crude product was dispersed in ethanol. The dispersion was centrifuged at 4800 r/min, washed with ethanol, and then collected using a membrane filter (1 μm) so as to give the product as a violet powder.

#### 3.1.2. Graphene Oxide (GO)

The graphene oxide was prepared by Hummer's method with a minor modification [21]. The mixture of graphite (2 g), grinded NaNO$_3$ (2 g), and H$_2$SO$_4$ (92 mL) was stirred for 30 min at 0 °C. Subsequently, KMnO$_4$ powder (10 g) was added carefully, and the resulting mixture was stirred at 35 °C for 60 min. Then, deionized water (92 mL) was dropped into the mixture slowly, and the mixture was heated at 95 °C for 20 min. Subsequently, deionized water (200 mL) was poured into the reaction mixture. Then, a 30% H$_2$O$_2$ solution (30 mL) was dropped very carefully so as to convert the manganese dioxide and unreacted permanganate into soluble sulfates in an ice bath. The mixture was centrifuged at 3000 r/min to remove the supernatant liquid. The precipitate was washed with a 5% HCl solution three times, and then with distilled water five times. The resulting solid was washed with ionized water three times, and exfoliated by ultrasonication for 2 h. The solution was centrifuged at 8000 r/min for 30 min, then the supernatant dispersion was centrifuged at 15000 r/min for 30 min to give the graphene oxide (GO) dispersion.

#### 3.1.3. GO–[Fe(Htrz)$_2$(trz)](BF$_4$) NRs Composite (**1**)

The mixture of GO in ethanol (30 mg/50 mL) and [Fe(Htrz)$_2$(trz)](BF$_4$) NRs in ethanol (60 mg/50 mL) was stirred at 25 °C for 6 h. After stirring, the brown product was centrifuged at 4000 r/min for 30 min, and the crude product was collected using a membrane filter (1 μm), washed with ethanol to give the product.

#### 3.1.4. rGO–[Fe(Htrz)$_2$(trz)](BF$_4$) NRs Composite (**2**)

Composite **1** was reduced to composite **2** by thermal treatments in a vacuum at 473 K for 12 h.

### 3.2. Measurement

All the measurements for composites **1** and **2** were performed three times using a film sample. Current-voltage (IV) properties were measured using an electrochemical analyzer, BAS, Model ALS/DY2323 BI-POTENTIOSTAT (Sumida-ku, Tokyo, Japan). The scanning electron microscopy (SEM) images and SEM-energy dispersive X-ray spectroscopy (SEM-EDX) data were collected on a JEOL, JSM-7600 F instrument (Akishima-shi, Tokyo, Japan). Fourier transform infrared spectra (FT-IR) were collected on SHIMADZU, IRAAffinity-1S (Kyoto-shi, Kyoto, Japan). The powder X-ray diffraction (PXRD) patterns were recorded on a Rigaku, MiniFlex II X-ray diffractometer (Akishima-shi, Tokyo, Japan). The temperature dependence of magnetic susceptibilities was measured on a Superconducting Quantum Interference Device (SQUID) magnetometer, Quantum Design Japan,

MPMSXL-5 (Toshima-ku, Tokyo, Japan). The samples were placed inside the SQUID chamber and measured between 300 and 400 K at field strengths of 0.5 T.

## 4. Conclusions

In summary, we have prepared a composite consisting of GO/rGO and a SCO complex of $[Fe(Htrz)_2(trz)](BF_4)$ NRs (30 × 200 nm). It was found that the shorter interlayer caused by the transformation of GO to rGO leads to pseudo-pressure effects. GO composite exhibited SCO behavior at $T_{1/2} = 357$ K, whereas for rGO composite $T_{1/2}$ was at 364 K. The observed pressure for $[Fe(Htrz)_2(trz)](BF_4)$ NRs estimated from the $T_{1/2}$ value corresponded to 24 MPa and being lower than that observed for the case of NPs (30 nm). Clearly, these findings provide new insight towards the research regarding the pressure effects in 2D materials including graphene, BN, $MoS_2$.

**Supplementary Materials:** The following are available online at http://www.mdpi.com/2312-7481/5/2/26/s1. Figure S1: I–V curves for **1** and **2**. Figure S2: SEM image of $[Fe(Htrz)_2(trz)](BF_4)$ NRs. Figure S3: FT-IR spectra for $[Fe(Htrz)_2(trz)](BF_4)$ NRs, composite **1,** and composite **2**. Figure S4: $\chi_m T$ vs. $T$ plot for $[Fe(Htrz)_2(trz)](BF_4)$ NRs. Figure S5: $\chi_g T$ vs $T$ plots for composite **1** and composite **2.**

**Author Contributions:** The manuscript was written through the contributions of all of the authors. All of the authors have given approval to the final version of the manuscript. The work was directed by S.H.

**Funding:** This work was supported by KAKENHI Grant-in-Aid for Scientific Research (A) JP17H01200.

**Conflicts of Interest:** The author declares no conflict of interest.

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
