# Peer review of "Pressure Effects with Incorporated Particle Size Dependency in Graphene Oxide Layers through Observing Spin Crossover Temperature"

_magnetochemistry, doi:10.3390/magnetochemistry5020026_

Reviewer 1 Report

In this study, the pressure effects with the incorporated particle size dependency in two-dimensional graphene oxide (GO) / reduced graphene oxide (rGO was evaluated in the interlayer spaces through observing the changes of the spin crossover (SCO) temperature (T1/2)

The manuscript presents an interesting investigation work, discussed interdisciplinary problematic which has not been presented in published scientific works. In my opinion, the manuscript is of the high scientific and engineering quality, and interesting for the readers of Journal of Magnetochemistry. The title confirms the content of the manuscript very well, abstract is correct, too. Taking all the previous into account, and regarding the overall high quality of the manuscript, I recommend it for publication in Journal of Magnetochemistry as an original scientific work.

It is recommended to apply the following comments:

- In Fig. 2 (c) and (D), the information (elements name) are not legible, authors need to use larger fonts

- Stats. method and number of test need to be included in the manuscript

Author Response

Q1.  In Fig. 2 (c) and (D), the information (elements name) are not legible, authors need to use larger fonts.

A1.  We have corrected the font size in Fig. (c) and (d).

Q2.  Stats. method and number of test need to be included in the manuscript.

A2.  We have added statements about status, method and number of tests in Measurement section.

Page 6, line 162.

Reviewer 2 Report

This is an interesting paper dealing with pressure effects upon the spin crossover T1/2 temperatures in 1D Fe triazole species caused by entrapment in graphene oxide and its reduced form.

The title must have spin crossover in it.

Shorthand NR and NP must be clearly defined

Is the crystal structure of [Fe(Htrz)2(trz)]BF4 known? to be able to related PXRD to simulated PXRD in these materials.

Mossbauer spectroscopy would help the characterisation of these composites

Overall, just enough new and significant to merit acceptance by Magnetochemistry

Author Response

Q1.  The title must have spin crossover in it.

A1.  We have changed the title.

“Pressure effects with incorporated particle size dependency in graphene oxide layers through observing spin crossover temperature”

Q2.  Shorthand NR and NP must be clearly defined

A2.  The NPs are spherical shape (30 nm ´ 30 nm) and the NRs are rectangle prism or cylinder shape (30 nm ´ 200 nm). We have described about that.

page 2, line 64, 67

Q3.  Is the crystal structure of [Fe(Htrz)2(trz)]BF4 known? to be able to related PXRD to simulated PXRD in these materials.

A3.  There are no reports about the crystal structure of [Fe(Htrz)2(trz)]BF4. Therefore, it is difficult to relate to simulated PXRD patterns obtained from crystal structure.

Q4.  Mossbauer spectroscopy would help the characterization of these composites

A4.  We carried out Mossbauer spectroscopy measurement for these composites. However, the spectra derived from [Fe(Htrz)2(trz)]BF4 NRs were not obtained because the composites each contained a small amount of iron(II) NRs.